# Epstein-Barr Virus Lytic Transcripts Correlate with the Degree of Myocardial Inflammation in Heart Failure Patients

**DOI:** 10.3390/ijms25115845

**Published:** 2024-05-28

**Authors:** Christian Baumeier, Dominik Harms, Britta Altmann, Ganna Aleshcheva, Gordon Wiegleb, Thomas Bock, Felicitas Escher, Heinz-Peter Schultheiss

**Affiliations:** 1Institute of Cardiac Diagnostics and Therapy, IKDT GmbH, 12203 Berlin, Germany; 2Robert Koch Institute, Unit 15: Viral Gastroenteritis and Hepatitis Pathogens and Enteroviruses, Department of Infectious Diseases, 13353 Berlin, Germany; 3Institute of Tropical Medicine, University of Tuebingen, 72074 Tuebingen, Germany; 4Deutsches Herzzentrum der Charité, Department of Cardiology, Angiology and Intensive Care Medicine, 12200 Berlin, Germany; 5DZHK (German Centre for Cardiovascular Research), Partner Site Berlin, 10785 Berlin, Germany

**Keywords:** endomyocardial biopsy, heart failure, myocarditis, inflammatory cardiomyopathy, Epstein-Barr virus, next-generation sequencing, viral transcripts

## Abstract

The Epstein-Barr virus (EBV) is frequently found in endomyocardial biopsies (EMBs) from patients with heart failure, but the detection of EBV-specific DNA has not been associated with progressive hemodynamic deterioration. In this paper, we investigate the use of targeted next-generation sequencing (NGS) to detect EBV transcripts and their correlation with myocardial inflammation in EBV-positive patients with heart failure with reduced ejection fraction (HFrEF). Forty-four HFrEF patients with positive EBV DNA detection and varying degrees of myocardial inflammation were selected. EBV-specific transcripts from EMBs were enriched using a custom hybridization capture-based workflow and, subsequently, sequenced by NGS. The short-read sequencing revealed the presence of EBV-specific transcripts in 17 patients, of which 11 had only latent EBV genes and 6 presented with lytic transcription. The immunohistochemical staining for CD3^+^ T lymphocytes showed a significant increase in the degree of myocardial inflammation in the presence of EBV lytic transcripts, suggesting a possible influence on the clinical course. These results imply the important role of EBV lytic transcripts in the pathogenesis of inflammatory heart disease and emphasize the applicability of targeted NGS in EMB diagnostics as a basis for specific treatment.

## 1. Introduction

Acute and chronic inflammations of the myocardium are often triggered by preceding or persisting viral infections and can lead to irreversible cardiac damage when the causative etiology is not rapidly identified and adequately treated. Although cardiac magnetic resonance imaging is widely used to verify acute myocardial inflammation, the analysis of viral involvement and low-grade chronic inflammatory processes requires the use of endomyocardial biopsies (EMBs). The recent analyses of EMBs from 871 patients with non-ischemic cardiomyopathy showed that the myocardial persistence of viral transcripts negatively affects long-term survival and the hemodynamic progression of cardiac function [1].

While virus detection has been restricted by polymerase chain reaction (PCR) limitations, metagenomic sequencing has gained attention in the advanced diagnostics of viral transcripts [2]. In contrast to conventional PCR methods, it is able to map viral transcripts across the entire genome and, therefore, enables the differentiation of latent and active viral infections.

The Epstein-Barr virus (EBV) is a human gammaherpesvirus that persists in the B lymphocytes of more than 90% of adults worldwide. EBV genomic DNA is commonly detected in the EMBs of heart failure patients [3], but the pathophysiologic role of the EBV in inflammatory cardiomyopathy remains poorly understood.

In this paper, we present a targeted next-generation sequencing (NGS) approach that enables the enrichment and detection of viral RNA transcripts in EMBs and investigate the relationship between EBV latent and lytic transcripts and the severity of myocardial inflammation in patients with heart failure with reduced ejection fraction (HFrEF).

## 2. Results

The baseline demographics and clinical characteristics of the study cohort are shown in Table 1. Most of the patients were male (59%), and the median age was 59 years, ranging from 18 to 82 years. All patients were White and of Caucasian ethnicity (Table 1). In all patients, EBV-specific DNA, but no other viral genome, was detected by quantitative PCR. LV function was impaired in all patients (LVEF = 30 ± 15%, Table 1), and the EMB analysis revealed varying degrees of lymphocytic inflammation (mean CD3^+^ T-cell count = 19 ± 16 cells/mm^2^). Active myocarditis was excluded according to the histopathologic Dallas criteria [4], and troponin values were normal in all patients (Table 1).

The enrichment and sequencing of the viral RNA libraries revealed EBV-specific transcripts in 17 of 44 patients (39%) (Figure 1a). In 11 of the 17 patients with sequenced EBV-specific transcripts, reads were mapped only to viral RNAs associated with upholding a latent infection (Lat^+^), including EBER-1/2 as well as the long non-coding BART transcripts RPMS-1 and A73 (Figure 2). Read mapping against viral lytic genes was observed in six patients, with five being additionally positive for latent genes (Lat^+^/Lyt^+^) (Figure 1b). The detected lytic genes are involved in the early viral transcriptional program, including DNA replication and the nuclear egress of the progeny virus, while the detected late genes encode for the structural components of EBV virions and the viral targets of anti-herpes drugs, BALF5, BGLF4, and BXLF1 (Figure 2).

The quantitative real-time PCR revealed no significant difference in viral DNA titer in all groups (Figure 3; no RNA: 187 ± 260 copies/μg DNA; Lat^+^: 678 ± 1600 copies/μg DNA; Lat^+^/Lyt^+^: 413 ± 574 copies/μg DNA; *p* = 0.274), suggesting that EBV transcript detection is independent of the genomic viral load.

Because EBV lytic genes are associated with the activation of an immune response [5], we next analyzed the myocardial immune cell infiltration. Indeed, we found a higher density of CD3^+^ T lymphocytes in EMBs of Lat^+^/Lyt^+^ (36 ± 23 cells/mm^2^) versus Lat^+^ (17 ± 10 cells/mm^2^, *p* = 0.048) and no RNA (15 ± 15 cells/mm^2^, *p* = 0.011) patients, suggesting an enhanced immune response in patients with an EBV lytic infection (Figure 4).

## 3. Discussion

EBV genomic DNA is commonly detected in the EMBs of patients with systolic LV dysfunction and might play a role in the pathogenesis of dilated cardiomyopathy (DCM) and even sudden cardiac death [3,6]. Therefore, an early diagnosis and treatment of the EBV infection is recommended to prevent DCM and congestive heart failure [7]. However, EBV DNA load seems not to contribute to cardiac dysfunction or inflammation, as it is frequently detected in patients without cardiac abnormalities [3]. Since EBV reactivation is involved in the pathogenesis of human malignancies and autoimmune diseases [5], the distinction between latent and lytic infections could help to assess the prognosis and specific therapy of EBV-positive heart failure patients. Conventional technologies, such as PCR analyses, are unsuitable for the detection of a large number of transcripts, as they only cover small areas of the virus genome. In this paper, we present a novel technique for the genome-wide detection of viral transcripts and show that the presence of EBV lytic transcripts correlates with the severity of myocardial inflammation and may therefore be of clinical relevance.

By using a targeted enrichment of EBV-specific RNA followed by NGS analysis, we were able to identify patients with the presence of EBV lytic transcripts. The detected lytic genes are mainly involved in the early transcriptional program for viral gene expression, DNA replication, the nuclear egress of the progeny virus, and the inhibition of host cell apoptosis during nascent virion production (Figure 2). Furthermore, they include late genes encoding for the structural components of nascent EBV virions and the viral targets of the acyclovir/ganciclovir treatment, BALF5, BGLF4, and BXLF1 (Figure 2). The transition from the latent to the lytic phase is necessary for EBV-mediated pathology and for the spread and persistence of the virus [5].

Previous studies suggest that the EBV infiltrates the myocardium not only via infected B cells, but possibly also via cardiomyocytes and the microvasculature [8,9]. As a consequence of cell death during the EBV lytic cycle in lymphocytes or cardiac cells, damage-associated molecular patterns (DAMPs) may lead to the stimulation of an immune response within the heart [10], and potentially to endothelial cell and vascular injury as previously shown in skin tissue [11]. Since inflammation can promote EBV reactivation [12] and the EBV lytic cycle may drive inflammation and cytokine production [13], it remains unclear whether the increased inflammation in EBV lytic patients is a cause or consequence of EBV reactivation. However, the parallel measurement of EBV transcripts and intramyocardial inflammation may be critical to determine a patient’s prognosis and optimal therapeutic strategy. Despite standard immunosuppressive therapy having shown favorable long-term outcomes in virus-negative myocarditis, the presence of viral genomes, particularly with active viral transcription, is contraindicated for such treatment [14]. Since an EBV lytic infection can be fatal in immunocompromised hosts, antiviral therapy should be considered in patients with evidence of EBV reactivation and concomitant inflammation. Although a number of antiviral agents inhibit chronically active EBV replication in vitro, their clinical success has been limited, and to date, no antiviral drug has been approved for the treatment of EBV infections [15]. Immunomodulatory agents, such as interferon (IFN)-beta, resulted in the almost complete clearance of the viral load in some cases of myocarditis with viral persistence [16] and suppressed a transcriptionally active parvovirus B19 infection in patients with viral cardiomyopathy [17]. For active herpesvirus infections, ganciclovir is suggested as a treatment option [18], and acyclovir is known for its inhibiting effects on EBV lytic replication in vitro [19]. However, the efficacy of the treatment with acyclovir/gancyclovir in patients with myocarditis or heart failure has not yet been proven and needs to be investigated in future clinical trials. Further attempts, such as anti-EBV-specific compounds, immunomodulatory agents, as well as cell therapies, might be promising for the treatment of chronically active EBV infections, as observed in a subset of HFrEF patients [15,20].

## 4. Materials and Methods

Patients: A total of 44 HFrEF patients (18 female; mean age of 58 (18–82) years) were enrolled in this study. Six to eight EMBs were taken from the left (N = 25) or right (N = 19) ventricle and examined for viral infections and myocardial inflammation. In all patients, coronary artery disease and other possible causes of myocardial dysfunction were excluded by angiography before the EMBs. Left ventricular ejection fraction (LVEF) was determined by echocardiography.

Histopathological inflammation diagnostics: To quantify the degree of myocardial inflammation, RNAlater-fixed, cryo-embedded EMBs were used for the immunohistochemical detection of CD3^+^ T cells [1]. According to the European Society of Cardiology (ESC) statement, myocardial inflammation was diagnosed by the detection of ≥7 CD3^+^ T cells per mm^2^ [21]. Three to five images of each EMB were taken and quantified by digital image analysis, as described before [1].

Molecular virology: RNAlater-fixed EMBs were used for genomic DNA and total RNA extraction, as previously described [1]. The detection of viral genomes, including Epstein-Barr virus (EBV), parvovirus B19 (B19V), enteroviruses (EV), adenoviruses (ADV), and human herpesvirus 6 (HHV-6), was performed using nested and quantitative PCRs [1]. All patients were positive for EBV-specific DNA and negative for the other viruses tested.

NGS detection of viral RNA transcripts: The experimental workflow is depicted in Figure 1b. The total RNA fraction was used for cDNA first- and second-strand synthesis, followed by sequencing library preparation using the NEBNext Ultra II FS DNA kit (New England Biolabs, Frankfurt, Germany). The viral libraries were enriched via the myBaits^®^ custom target capture kit (Daicel Arbor Biosciences, Ann Arbor, MI, USA) and sequenced on an Illumina MiSeq^TM^ (2 × 150 bp paired-end run; 5 × 10^5^ reads/sample; 300 cycles v2 reagent kit; San Diego, USA). Read processing was performed with Trimmomatic (version 0.39). Paired reads were taxonomically classified via Kraken2 with a confidence threshold of ≥10 reads and mapped against EBV reference sequences (Figure 2; NC_007605.1 and NC_009334.1) using HiSat2 with deduplication by Samtools markdup. The sequence coverage of viral genes was analyzed by QualiMap (version 2.2.2c) and visualized with Geneious Prime 2023.2.1 using deposited annotations for EBV genes and spliced transcripts.

Statistics: A one-way ANOVA followed by Tukey’s post hoc test was used to compare the differences among the three groups. All statistical analyses were performed using GraphPad Prism version 9.4.1.

## 5. Conclusions

Our work highlights metagenomic NGS as a powerful tool in EMB diagnostics and provides the first evidence of an association between EBV reactivation and the degree of myocardial inflammation in patients with heart failure. Although these results are limited because of their preliminary nature and need to be confirmed in future clinical trials, they could have strong implications for risk stratification and treatment decisions in EBV-positive patients with inflammatory heart disease.

## Figures and Tables

**Figure 1 ijms-25-05845-f001:**
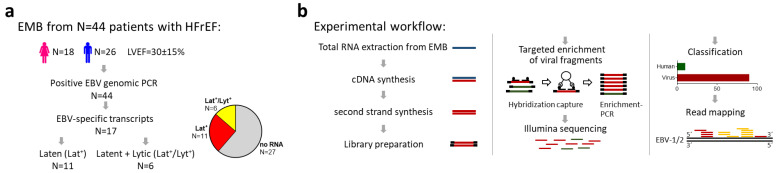
(**a**) Schematic illustration of the analyzed cohort of N = 44 patients with heart failure with reduced ejection fraction (HFrEF) and Epstein-Barr virus (EBV)-positive genomic PCR. EBV-specific transcripts were found in the RNA fractions of endomyocardial biopsies (EMBs) in 17 of 44 patients. Of these, 11 patients had transcripts from latent (Lat^+^) and 6 patients from latent and lytic genes (Lat^+^/Lyt^+^). (**b**) Experimental workflow for the enrichment and detection of viral transcripts in EMBs.

**Figure 2 ijms-25-05845-f002:**
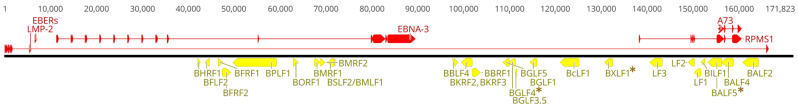
Genomic locations of viral latent (red) and lytic (yellow) transcripts are indicated in the EBV reference genome. Viral targets of acyclovir/ganciclovir are marked with asterisks.

**Figure 3 ijms-25-05845-f003:**
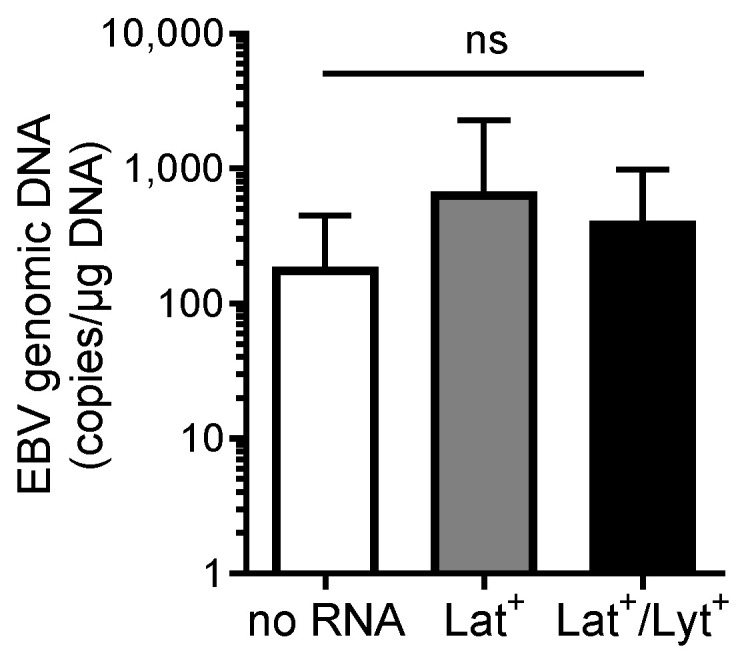
EBV genomic DNA titers (expressed as EBV DNA copies per μg total DNA) in EMBs from patients without the detection of EBV-specific RNA (no RNA) and patients with the detection of latent (Lat^+^) or latent and lytic RNA (Lat^+^/Lyt^+^). Data are represented as mean ± SD. ns, not significant.

**Figure 4 ijms-25-05845-f004:**
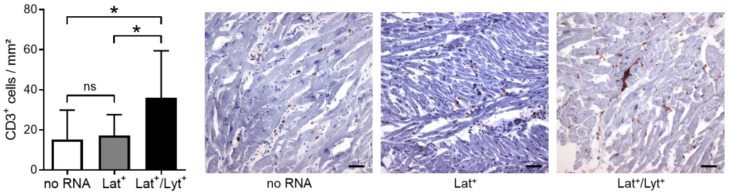
Representative immunostaining and quantification of CD3^+^ T lymphocytes in EMB cryo-sections of patients without the detection of EBV-specific RNA (no RNA) and patients with the detection of latent (Lat^+^) or latent and lytic RNA (Lat^+^/Lyt^+^). Magnification: 200×. Scale bars: 50 μm. Data are represented as mean ± SD. * *p* < 0.05; ns, not significant.

**Table 1 ijms-25-05845-t001:** Baseline demographics and characteristics of study cohort.

Parameter	Number (%)
Sex	
Female	18 (41)
Male	26 (59)
Age	
Median (years)	59
Range (years)	18–82
Ethnicity	
Caucasian	44 (100)
LVEF (%)	30 ± 15
Troponin (pg/mL)	<15

## Data Availability

All authors confirm that all related data supporting the findings of this study are presented in the article.

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
