# Peer review of "Epstein-Barr Virus Lytic Transcripts Correlate with the Degree of Myocardial Inflammation in Heart Failure Patients"

_ijms, 2024, doi:10.3390/ijms25115845_

Round 1

Reviewer 1 Report

Comments and Suggestions for Authors

The study entitled "Epstein-Barr Virus Lytic Transcripts Correlate with Degree of Myocardial Inflammation in Heart Failure Patients" examines the relationship between Epstein-Barr virus (EBV) lytic transcripts and myocardial inflammation in patients with heart failure with reduced ejection fraction (HFrEF). Using targeted next-generation sequencing (NGS), the study investigates the presence of EBV viral transcripts in 44 HFrEF patients who have tested positive for EBV DNA.

The research method involves the enrichment of EBV-specific transcripts from endomyocardial biopsies (EMB) using a customized hybridization-based workflow, followed by sequencing by NGS. This approach allows a detailed investigation of viral activity in cardiac muscle tissue.

Immunohistochemical analysis was performed to stain CD3+ T lymphocytes, a marker of immune response in cardiac tissue. The results indicate a significant increase in myocardial inflammation that correlates with the presence of lytic EBV transcripts. This finding suggests that active EBV infection may influence the clinical course of heart failure in these patients by exacerbating the inflammatory processes.

The study highlights the potential of using NGS to identify viral components that may influence the severity of myocardial inflammation and provides insights into how EBV may alter clinical outcomes in patients with HFrEF. These findings could pave the way for targeted therapeutic strategies that address viral influences on heart failure pathology.

How do the results of this study contribute to our understanding of heart failure pathogenesis?

What are the implications of the results of this study for the development of potential therapeutic strategies targeting EBV in patients with heart failure?

What specific NGS techniques were used to sequence EBV-specific transcripts in the endomyocardial biopsies, and how did these techniques contribute to the identification of latent and lytic EBV genes in the samples?

Author Response

First of all, we would like to thank reviewer 1 for his/her helpful and constructive questions which we have tried to answer.

How do the results of this study contribute to our understanding of heart failure pathogenesis?

Our data suggest an important impact of active (lytic) EBV infection on intramyocardial inflammation in patients with heart failure. It is known that viral infection is the most common trigger for myocarditis and that viral persistence can lead to chronic heart failure. We therefore assume that an active EBV infection leads to damage to the infected cells in the heart muscle and thus to an inflammatory response. As a result of the chronic inflammation, the heart tissue is damaged, leading to heart failure.

Without being able to prove this correlation on the basis of the available data, we believe that this finding may play a key role in understanding the pathophysiology of heart failure.

What are the implications of the results of this study for the development of potential therapeutic strategies targeting EBV in patients with heart failure?

To date, there are no guideline-based specific treatment recommendations for patients with EBV infections. Although a number of antiviral agents inhibit chronically active EBV replication in vitro, their success in the clinic is limited.

Our results suggest that a specific subgroup of EBV-positive HFrEF patients may benefit from antiviral therapy. In this context, anti-EBV therapeutics targeting lytic replication are of high clinical importance. In the past decades, antiviral agents (including acyclovir, ganciclovir and vidarabine), immunomodulatory agents (such as interferon-α and interleukin-2), chemotherapeutic agents and cell therapies have been used to treat active EBV infections, but with limited success. The further development of EBV-specific therapeutic strategies could contribute to improving the prognosis of HFrEF patients with lytic EBV replication.

We have tried to go into this in more detail in the revised version of the discussion.

What specific NGS techniques were used to sequence EBV-specific transcripts in the endomyocardial biopsies, and how did these techniques contribute to the identification of latent and lytic EBV genes in the samples?

After enrichment of the EBV-specific fragments with the myBaits® custom target capture kit from Arbor Biosciences, we used paired-end sequencing on an Illumina MiSeqTM (2×150bp paired-end run; 5×105 reads/sample) using the v2 300 cycles reagent kit. We specify this in the material and methods section.

This paired-end sequencing yields RNA-fragments of around 150 bp that can be assigned to the EBV genome. The bioinformatic alignment of these sequences to the reference genome using QualiMap and Genious Prime enables the assignment of the transcripts and spliced variants to latent and lytic EBV genes.

Reviewer 2 Report

Comments and Suggestions for Authors

This is an interesting report about the impact of EBV gene for the myocardial inflammation. They showed the presence of EBV-specific transcripts in 17 patients, of which 11 had only latent EBV genes, and 6 presenting with lytic transcription. Immunohistochemical staining for CD3+ T lymphocytes showed a significant increase in the degree of myocardial inflammation in the presence of lytic EBV transcripts. I have several comments for the manuscript.

# Do you have any information about parameters of EBV in blood samples?

# Baseline demographics of study patients should be presented.

# How about any differences in BNP or troponin?

# How about the association between myocardial inflammation and clinical outcomes such as heart failure rehospitalization?

Comments on the Quality of English Language

No comment

Author Response

Thank you for the constructive comments/suggestions and for providing confidence to our work. We tried to follow your suggestions as follows.

Do you have any information about parameters of EBV in blood samples?

We analyzed blood samples from N=11 patients and found a positive result for genomic EBV DNA in all of them. We did not have access to blood samples from all other patients and could not detect any systemic EBV load. Therefore, we can only speculate about EBV parameters in the blood.

However, it is known that virus serology and positive PCR results from peripheral blood have not proven useful for the accurate detection of the causative pathogen in the myocardium [1]. The detection of EBV in the blood therefore only allows very limited conclusions to be drawn about the infection of the myocardium and has therefore only been carried out in a few cases.

  1. Caforio, A.L.P.; Pankuweit, S.; Arbustini, E.; Basso, C.; Gimeno-Blanes, J.; Felix, S.B.; Fu, M.; Helio, T.; Heymans, S.; Jahns, R.; et al. Current State of Knowledge on Aetiology, Diagnosis, Management, and Therapy of Myocarditis: A Position Statement of the European Society of Cardiology Working Group on Myocardial and Pericardial Diseases. Eur. Heart J. 2013, 34, 2636–2648, doi:10.1093/eurheartj/eht210.

Baseline demographics of study patients should be presented.

Many thanks for this hint. We present baseline demographics in the new table 1.

How about any differences in BNP or troponin?

As requested by the reviewer, we added troponin values. As these are not acute myocarditis patients but chronic cardiomyopathy patients, the troponin values were not increased, as expected. We describe this in the revised manuscript.

Unfortunately, we do not have data on BNP for all patients. Due to the incompleteness of these data we did not include the BNP data in the manuscript.

How about the association between myocardial inflammation and clinical outcomes such as heart failure rehospitalization?

This is an interesting question, which unfortunately we cannot answer definitively. We have no clinical follow-up of these patients and cannot say much about the course of heart failure. We know from other studies that increased intramyocardial inflammation is associated with worsened cardiac function and a higher rate of rehospitalization. It can therefore also be assumed here that Lat+/Lyt+ patients have a worse clinical outcome due to an increased inflammation.

Round 2

Reviewer 1 Report

Comments and Suggestions for Authors

the authors responded to my comments.

Reviewer 2 Report

Comments and Suggestions for Authors

The revised manuscript was finely corrected.

Comments on the Quality of English Language

No comment